# Evaluating the Efficacy of Water-Soluble Bone Wax (Tableau Wax) in Reducing Blood Loss in Spinal Fusion Surgery: A Randomized, Controlled, Pilot Study

**DOI:** 10.3390/medicina59091545

**Published:** 2023-08-25

**Authors:** Jung Guel Kim, Dae-Woong Ham, Haolin Zheng, Ohsang Kwon, Ho-Joong Kim

**Affiliations:** 1Spine Center and Department of Orthopedic Surgery, Seoul National University College of Medicine and Seoul National University Bundang Hospital, 166 Gumiro, Bundang-gu, Sungnam 463-707, Republic of Korea; 54482@snubh.org (J.G.K.); haolin0622@gmail.com (H.Z.); ormssang@gmail.com (O.K.); 2Department of Orthopedic Surgery, Chung-Ang University Hospital, College of Medicine, Chung-Ang University, 102, Heukseok-ro, Seoul 06973, Republic of Korea; hamdgogo@gmail.com

**Keywords:** water-soluble bone wax, hemostasis, lumbar interbody fusion, spine surgery

## Abstract

*Background and Objectives*: Lumbar decompression with fusion surgery is an effective treatment for spinal stenosis, but critical postoperative hematoma is a concern. Bone wax has been widely used to control bone bleeding but it has some drawbacks. This study aimed to evaluate the efficacy of Tableau wax, a bioabsorbable hemostatic material, in patients undergoing spinal fusion surgery through a pilot study design. *Materials and Methods*: A total of 31 patients were enrolled in this single-surgeon, single-institution study. The participants underwent transforaminal lumbar interbody fusion surgery and were randomly assigned to the control group (Bone wax) or test group (Tableau wax). Demographic data, pre- and post-operative hemoglobin levels, blood loss volume, surgical time, Oswestry Disability Index, and EQ-5D scores were recorded. *Results*: The study showed no significant difference in preoperative and postoperative hemoglobin levels, Oswestry Disability Index, and EQ-5D scores between the groups. However, the Tableau wax group had a significantly lower reduction in hemoglobin levels (1.3 ± 1.0 g/dL) and blood loss (438.2 mL) compared to the Bone wax group (2.2 ± 0.9 g/dL and 663.1 mL, respectively; *p* = 0.018 and *p* = 0.022).

## 1. Introduction

Lumbar decompression, with or without fusion surgery, is an effective surgical option for patients suffering from spinal stenosis [1]. However, since osteotomy procedures are an integral part of the surgery, the incidence of critical postoperative hematoma has been reported to range from 0.1 to 1.0% [2,3]. Consequently, implementing safe and efficient hemostasis methods is crucial for minimizing bleeding during spinal surgery.

Bone wax, a hemostatic material containing beeswax (J&J, New Brunswick, NJ, USA), has been extensively used to prevent bone bleeding during surgery [4,5]. Its main component, paraffin wax, is a non-degradable material in the human body that remains permanently at the application site. In some reports, residual wax has been associated with inhibiting normal bone formation and potentially leading to infection or granulation tissue formation [4,6,7].

To address these concerns, Tableau wax, a bone hemostatic material utilizing a bioabsorbable PEO-PPO copolymer (polyethylene oxide-polypropylene oxide copolymer, Poloxamer), has been developed and implemented in clinical practice [8]. Tableau wax not only acts as a physical barrier when applied to the bone bleeding site, but also offers the advantage of not hindering bone formation due to its absorption into the human body.

Although the clinical usefulness of water-soluble bone wax materials has been reported in several studies [9,10,11], its efficacy in spine surgery remains less well-established. Therefore, this study aims to evaluate the effectiveness of Tableau wax on bone hemostasis in patients undergoing spinal fusion surgery through a pilot study. This investigation will provide valuable insights into the potential benefits and safety of using Tableau wax as a hemostatic agent in spinal surgery settings.

## 2. Materials and Methods

This pilot study was approved by our hospital institutional review board. We collected patients’ basic demographic data, including age, sex, height, weight, surgical time, and pre- and post-operative hemoglobin (Hb) levels. The inclusion criteria were as follows: (1) age over 19 years and (2) underwent transforaminal lumbar interbody fusion (TLIF) surgery for spinal stenosis at a single level. The exclusion criteria were as follows: (1) patients with a bleeding tendency due to coagulation disorders such as ongoing anticoagulation therapy or idiopathic thrombocytopenic purpura, (2) patients who need to use anticoagulants for reasons other than spinal diseases such as cardiovascular stent, and (3) patients with an immunologic disorder or hypersensitivity-related disease. From September 2020 to April 2021, 32 consecutive patients who underwent lumbar fusion surgery for spinal stenosis were included. Of these, one patient was excluded. Finally, a total of 31 patients were enrolled.

### Surgical Procedure

Patients were randomly assigned into the control group and test group using a simple equal probability randomization scheme. Patients underwent a standard open single-level TLIF procedure with pedicle screws, morselized local bone grafts, and interbody cages. Local bone grafts were obtained bilaterally from the spinous processes, lamina, and facets using surgical tools and bony decompression was achieved simultaneously. These bony decompressed sites were considered to generate major focuses of bone bleeding. Thus, the patients assigned to the control group were applied bone wax to control the bone bleeding during the surgery, while the patients in the test group were applied Tableau wax.

The surgical technique used followed a standard procedure. Patients were placed in the prone position and underwent general anesthesia. Coagulation optimization, such as IV or local tranexamic acid injection, was not performed on patients. A midline incision was made, and the posterior elements of the spine were exposed. The lamina and facet joints were removed by Kerrison rongeur and punch forceps, and high speed burr to access the intervertebral disc. The disc was then removed, and the endplates were prepared for fusion. In the test group, an osteotomy was performed to create the appropriate angle for the interbody cage insertion. After osteotomy, Tableau wax was applied to the bleeding site to stop the bleeding. A transforaminal approach was used to insert a cage into the intervertebral space. A single, 4.8 mm diameter sized closed drainage suction system (Hemovac) was inserted for proper drainage of postoperative bleeding in the patient in each case. In the control group, the majority of surgical procedures were conducted in the same manner as the test group, as described above. The only difference was that bony bleeding was controlled by the application of conventional bone wax instead of Tableau wax in the control group. Postoperative care was standardized for all patients, and they were encouraged to mobilize as soon as possible after surgery. Follow-up visits were scheduled at regular intervals to monitor clinical outcomes, radiographic fusion, and any adverse events.

The primary endpoint for efficacy in bleeding control was perioperative blood loss, calculated according to the following formula: total body volume (TBV), a formula by Nadler et al.
Male: TBV [mL] = (0.0003669 × height^3^ [cm]) + (32.19 × body weight [kg]) + 604;
Female: TBV [mL] = (0.0003561 × height^3^ [cm]) + (33.08 × body weight [kg]) + 183.
Perioperative blood loss [mL] = TBV [mL] × (Hbi − Hbe)/Hbi + Sum of blood products transfused [mL]
(Hbi [g/dL]: preoperative Hb level, Hbe [g/dL]: postoperative Hb level).

## 3. Results

The study included a total of 31 patients, with 15 patients in the bone wax group and 16 patients in the Tableau wax group. The patients’ demographics, including age, gender, and BMI, were comparable between the two groups (Table 1). The mean age was 67.7 ± 6.7 years in the bone wax group and 67.8 ± 9.8 years in the Tableau wax group (*p* = 0.996). The proportion of female patients was similar in both groups, with 66.7% (N = 10) in the bone wax group and 62.5% (N = 10) in the Tableau wax group (*p* > 0.99). The mean BMI was 23.9 ± 2.5 kg/cm^2^ for the bone wax group and 25.1 ± 2.3 kg/cm^2^ for the Tableau wax group (*p* = 0.156). Regarding the fusion levels, there was no significant difference between the groups (*p* = 0.750). The distribution of fusion levels for L2-3, L3-4, L4-5, and L5-S1 was 6.7%, 6.7%, 73.3%, and 13.3% in the bone wax group, and 0%, 6.3%, 75.0%, and 18.8% in the Tableau wax group, respectively.

The Oswestry Disability Index (ODI) and EQ-5D scores were evaluated preoperatively and 12 months postoperatively (Table 1). There was no significant differences between the two groups for both preoperative ODI (0.519 ± 0.176 for the bone wax group and 0.492 ± 0.200 for the Tableau wax group, *p* = 0.696) and postoperative ODI at 12 months (0.177 ± 0.145 for the bone wax group and 0.179 ± 0.130 for the Tableau wax group, *p* = 0.937). Similarly, the preoperative EQ-5D scores showed no significant difference (0.372 ± 0.195 for the bone wax group and 0.425 ± 0.230 for the Tableau wax group, *p* = 0.495). The postoperative EQ-5D scores at 12 months also showed no statistically significant difference between the two groups (0.798 ± 0.101 for the bone wax group and 0.690 ± 0.239 for the Tableau wax group, *p* = 0.113).

Table 2 shows the preoperative and postoperative hematologic characteristics of both groups. The operation time between the two groups showed no significant difference (923.5 ± 181.7 min for the bone wax group and 943.1 ± 142.5 min for the Tableau wax group, *p* = 0.740). The preoperative hemoglobin (Hb) levels were similar between the two groups (13.4 ± 1.6 g/dL for the bone wax group and 13.0 ± 1.4 g/dL for the Tableau wax group, *p* = 0.478). The postoperative Hb levels were also comparable (11.2 ± 1.4 g/dL for the bone wax group and 11.6 ± 1.3 g/dL for the Tableau wax group, *p* = 0.326). However, the Hb reduction was significantly lower in the Tableau wax group (1.3 ± 1.0 g/dL) compared to the bone wax group (2.2 ± 0.9 g/dL, *p* = 0.018). Furthermore, the blood loss was significantly less in the Tableau wax group (438.2 mL) compared to the bone wax group (663.1 mL, *p* = 0.022). The incidence of incidental durotomy was observed in two cases, with one case occurring in the bone wax group and one case in the Tableau wax group. In the bone wax group, there was one instance of superficial surgical site infection, which was successfully treated with antibiotics without the need for further surgery. Neither group experienced complications, such as surgery-related hematoma or the need for revision surgery.

## 4. Discussion

Major blood loss in spinal fusion comes from the osteotomy procedure [12]. The removal of osteophytes and blood exudation from the bone section or screw hole is an important source of blood loss in which the control of bleeding is not quite satisfactory using electrocauterization or ligature control. Hence, the direct application of hemostatic agents on the exposed bone has been gaining considerable interest.

The effectiveness of bone wax in reducing blood loss has been previously proved through studies, due to its ability to immediately seal and plug the bone marrow sinusoids and consequently prevent the oozing of blood [5]. The results presented in Table 2 indicate that Tableau wax demonstrates better hemostatic capabilities than conventional bone wax in single-level lumbar fusion surgery. One possible reason for the superior hemostatic ability of Tableau wax could be its unique formulation, which may provide more effective sealing and clotting properties when applied to bleeding bone surfaces. This is supported by the observation that Hb reduction and blood loss were both significantly lower in the Tableau wax group compared to the bone wax group, despite similar preoperative and postoperative Hb levels in both groups. Additionally, Tableau wax may adhere more effectively to the bone surface, creating a more efficient physical barrier to blood flow, which may contribute to reduced blood loss. The properties of Tableau wax, such as its consistency, ease of application, and ability to remain in place during surgery, could also play a role in enhancing hemostasis compared to bone wax. Further research is needed to better understand the specific mechanisms behind the improved hemostatic performance of Tableau wax in spinal surgery and to evaluate the potential clinical implications of these findings.

One of the significant factors that is considered in successful spinal surgery is the rate of bone union [1,13]. Bleeding focuses, such as osteotomy sites and pits drilled for pedicle screws, are those requiring bone wax application. Meanwhile, these are also potential sites that may have impact on bone genesis. We believe that the bioabsorbable Tableau wax should be a promising material through its minimal interference, but its efficacy regarding bone formation was not reported in the current pilot study. Since the study was designed to compare the hemostatic efficacy of different bone wax, data related to bony fusion rate, such as the patients’ post-op CT, were not acquired. Nevertheless, since critical sites of bony union, those in intervertebral spaces, were spared from bone wax application, we assumed that both types of waxes may have minimal impact on fusion rates for our cases, in which patients all received single-level transforaminal lumbar interbody fusion (TLIF) surgery. But the true impact of Tableau wax on the fusion rates of spinal surgery may, indeed, require a longer-term study in order to be statistically analyzed.

With a few positive features shown in the Tableau wax, the usage of the foreign agent would still possess the concern of infection and the application of a hemostatic agent which can potentially permeate into the systemic circulation which may lead to thromboembolism events [4,7,14]. Infection and thromboembolic events are important considerations because of the serious consequences for the outcome of elective orthopedic surgery. Since there are reported complications for the application of the bone wax, further study with bigger sample sizes will be required to draw conclusions on the safety of Tableau wax.

The strengths of this study are multi-faceted. To the authors’ knowledge, this may be the only clinical trial that evaluates the effectiveness of bioabsorbable bone wax in spinal fusion surgery. The single-surgeon design ensured a uniform method of applying the bone wax and standardized surgical technique, and the single-institution design secured a uniform protocol for postoperative care that collectively minimized systematic variables for the study. Lastly, the perioperative blood loss was determined using the hemoglobin balance method [15]. Visually estimating blood loss by clinically examining the amount of blood collected in the suction canisters, surgical sponges, drapes, and towels can underestimate the actual blood loss [16]. The amount of postoperative drainage is not an accurate reflection of total blood loss as not all blood is evacuated by the drain. There may be significant hidden blood loss due to bleeding into the tissues and residual blood in the joint [16].

Meanwhile, this study does have limitations. As mentioned earlier, the safety of Tableau wax in regards to complications were not reported [4]. Secondly, we chose only candidates for single-level spinal fusion. In general, 1- or 2-level TLIF surgeries may not typically result in significant bleeding at the osteotomy site, and the use of bone wax is not commonly necessary for hemostasis. There would be a considerable increase in perioperative blood loss in multi-level spinal fusion surgery, especially in cases where patients have a tendency to bleed. However, this study aimed to create an environment that minimizes potential biases, such as different surgical level and patient factors, and specifically focused on evaluating the hemostatic properties of water-soluble bone wax. Hence, whether Tableau wax would show the same superior hemostasis function in multi-level spinal fusion surgery is yet to be identified. Our clinical pilot study may have limitations in further understanding the natural properties and biological role of Tableau wax that supposedly propel its superiority over conventional bone wax. Additional in vivo studies shall be required to understand the precise mechanism of this bioabsorbable wax, which may also help distinguish the safety of the material. Lastly, only patients free of any coagulation disorders, and not taking any anti-coagulants due to their underlying past medical conditions, were included [17]. We believe these limitations have not impacted the primary result.

## 5. Conclusions

Tableau wax showed satisfactory bleeding control ability compared to conventional bone wax. Therefore, since Tableau wax has advantages in biological properties, it can be effectively applied for bone bleeding control in lumbar fusion surgery.

## Figures and Tables

**Table 1 medicina-59-01545-t001:** Descriptive analysis of the demographic parameters.

	Bone Wax	Tableau Wax	Total	*p*
Patients number N	15	16	31	
Age (years)	67.7 ± 6.7	67.8 ± 9.8	67.7 ± 8.3	0.996
Female, N (%)	10 (66.7%)	10 (62.5%)	20 (64.5%)	1.000
BMI (kg/cm^2^)	23.9 ± 2.5	25.1 ± 2.3	24.5 ± 2.4	0.156
Fusion level, N (%)				0.750
L2-3	1 (6.7)	0	1 (3.2)	
L3-4	1 (6.7)	1 (6.3)	2 (6.5)	
L4-5	11 (73.3)	12 (75.0)	23 (74.2)	
L5-S1	2 (13.3)	3 (18.8)	5 (16.1)	
ODI				
Preoperative	0.519 ± 0.176	0.492 ± 0.200	0.505 ± 0.186	0.696
Posteoperative 12 months	0.177 ± 0.145	0.179 ± 0.130	0.178 ± 0.135	0.937
EQ-5D				
Preoperative	0.372 ± 0.195	0.425 ± 0.230	0.400 ± 0.212	0.495
Posteoperative 12 months	0.798 ± 0.101	0.690 ± 0.239	0.742 ± 0.191	0.113

**Table 2 medicina-59-01545-t002:** Comparison of operation time, hemoglobin levels, transfusion volume, hemoglobin reduction, and blood loss between the two groups.

	Bone Wax	Tableau Wax	Total	*p*
Operation time (min)	923.5 ± 181.7	943.1 ± 142.5	933.6 ± 160.2	0.740
Preoperative Hb	13.4 ± 1.6	13.0 ± 1.4	13.2 ± 1.5	0.478
Postoperative Hb	11.2 ± 1.4	11.6 ± 1.3	11.4 ± 1.3	0.326
Transfusion (mL)	150 ± 141.42	166.67 ± 104.08	160.0 ± 102.5	0.887
Hb reduction (mL)	2.2 ± 0.9	1.3 ± 1.0	1.7 ± 1.0	**0.018**
Blood loss (mL)	663.1 ± 240.0	438.2 ± 273.6	548.7 ± 278.8	**0.022**

Bold values indicate *p* < 0.05.

## Data Availability

Data to support the findings of this study are available upon reasonable request.

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
