# Peer review of "Evaluating the Efficacy of Water-Soluble Bone Wax (Tableau Wax) in Reducing Blood Loss in Spinal Fusion Surgery: A Randomized, Controlled, Pilot Study"

_medicina, 2023, doi:10.3390/medicina59091545_

Round 1
Reviewer 1 Report (Previous Reviewer 1)
Authors present their re-submitted manuscript on randomized controlled pilot single surgeon single institution study which involved 31 patients who underwent TLIF surgery to investigate if control group (use of Bone wax for bone bleeding) or test group (water soluble Tableau wax) had comparable outcomes. The authors have adressed most of the reviewer remarks from the previously reject/resubmit manuscript. However, in the conclusions I would sincerely advise to add the remarks from the discussion - in mono/bisegmental TLIF surgery, which is NOT a very "bloody" surgery, Tableu wax has shown similar Hb-reduction and transfusion need as conventional wax - however, the true question is - what about TLIF cases where there is no wax at all? I suggest to include the historical cohort from the same surgeon without any use of wax for comparison.
Acceptable.
Author Response
Reviewer #1.
Comment:
Authors present their re-submitted manuscript on randomized controlled pilot single surgeon single institution study which involved 31 patients who underwent TLIF surgery to investigate if control group (use of Bone wax for bone bleeding) or test group (water soluble Tableau wax) had comparable outcomes.
The authors have adressed most of the reviewer remarks from the previously reject/resubmit manuscript.
However, in the conclusions I would sincerely advise to add the remarks from the discussion - in mono/bisegmental TLIF surgery, which is NOT a very "bloody" surgery, Tableu wax has shown similar Hb-reduction and transfusion need as conventional wax - however, the true question is - what about TLIF cases where there is no wax at all?
I suggest to include the historical cohort from the same surgeon without any use of wax for comparison.
Response:
Thank you for your insightful comment.
In response to your suggestion, we examined blood loss in a cohort of 16 patients who underwent single-level lumbar fusion surgery without the use of bone wax. These patients had similar demographic characteristics and no coagulation disorders. The mean total blood loss for this group was 717.4 ± 276.3 ml. While this value was higher than that observed in patients using conventional bone wax, the difference was not statistically significant (P = 0.566).
We concur with your perspective that presenting data from a bone-wax-free control group can offer a more comprehensive understanding for our readers. However, given that our study was initially designed as a randomized preliminary investigation, there are inherent limitations in retrospectively analyzing and presenting cohort data as a control group. We plan to address this in a subsequent study, focusing on the wax's clinical effectiveness in multilevel surgeries. We hope this addresses your concerns and appreciate your valuable feedback.
Reviewer 2 Report (Previous Reviewer 2)
Dear authors, I commend your efforts to analyze the effectiveness of using a bioabsorbable bone wax in the use of TLIF surgery but my concerns as rightly pointed out in the limitations that its use could be best evaluated and brought out in multilevel degenerative surgeries or deformity correction surgery where it is commonly used. How does the author validate his results to be generalized for the use in rest of the surgery types
minor language errors and conclusion needs rephrasing for clear message to be conveyed to the readers
Author Response
Comment:
Dear authors, I commend your efforts to analyze the effectiveness of using a bioabsorbable bone wax in the use of TLIF surgery but my concerns as rightly pointed out in the limitations that its use could be best evaluated and brought out in multilevel degenerative surgeries or deformity correction surgery where it is commonly used. How does the author validate his results to be generalized for the use in rest of the surgery types
Response:
Thank you for your insightful feedback.
You're right in highlighting the importance of evaluating the effectiveness of bioabsorbable bone wax in surgeries where its use is more common, such as multilevel degenerative surgeries or deformity correction surgery. While our study primarily focused on single-level fusion surgery, we recognize the need to extrapolate our findings to more complex surgical scenarios.
In our cohort of single-level lumbar fusion surgeries, we assessed blood loss in 16 patients who did not use bone wax. These patients had no coagulation disorders and shared similar demographic characteristics. The average total blood loss for this group was 717.4 ± 276.3 ml. Even though this was higher than in patients using conventional bone wax, the difference wasn't statistically significant.
While presenting this control group would have been ideal, the design of our study as a randomized preliminary investigation poses challenges in retrospectively analyzing and presenting this data. We plan to address this in a subsequent study, where we will focus on the clinical effectiveness of the wax in multilevel surgeries. We hope this response addresses your concerns and appreciate your valuable input.
Round 2
Reviewer 1 Report (Previous Reviewer 1)
Authors have sufficiently responded to remarks.
Ok.
This manuscript is a resubmission of an earlier submission. The following is a list of the peer review reports and author responses from that submission.
Round 1
Reviewer 1 Report
Authors present a randomized control trial, i.e. single surgeon prospective study on 31 patients who received transforaminal lumbar interbody fusion (TLIF) and who were randomly assigned to the control group (Bone wax) or test group (Tableau wax). Tableau wax group had a significantly lower reduction in hemoglobin levels and blood loss compared to bone vax group.
Low number of patients is limitation of the study; since this is a single surgeon study, reproducibility is not possible. It is more a prospective pilot study then a randomized control trial. Did any of the patients use blood thinners or had impaired coagulation? Results section are too short. I suggest to include the table with all patients, age, gender, type of surgery and very important - duration of surgery. Were this all one -level cases? It is also unclear in which situation was the bone wax i.e. Tableu used - many spine surgeons operate 1- and 2-level TLIF without experienceg any major bleeding and without need to use bone wax at all. It is generally unusual to use it in a way where you would expect a significant effect on hemostasis and blood loss. Please provide detailed explanation and illustrative cases. Did any patient underwent substitution or any kind of optimization of coagulation? What is the complication rate? Screw positions? Durotomy incidence? Did you put drains and did you calculate amount of blood in the drain and added it to blood loss?
Minor editing requiered.
Reviewer 2 Report
Dear authors
I congratulate you on making this RCT that compares the use of bioabsorbable wax to traditional wax in spine surgery. However, there are some major flaws in the methodology of the paper that prevents me from considering your work for publication.
1. Registration data of this RCT is missing
2. Patient flow diagram of RCT as per reporting guidelines is missing
3. The control group interventions are not detailed
4. The clinical significance of the difference noted need to be discussed
5. The biological role of the wax saying that its bioabsorbable is not being tested or its fusion rates
6. If the difference between the two groups of wax is only on its bioabsorbable nature how do the authors find a difference between the two wax used in the short-term outcomes
7. the design of the study does not test the natural properties of the wax that is claimed over the routinely used wax
Minor language errors that need rectification with the help of a native language speaker